



# Quality assurance and control on hydrological data off western Sardinia (2000 - 2004), western Mediterranean

Alberto Ribotti[1], Roberto Sorgente[1], Mireno Borghini[2]

[1]Istituto per lo studio degli impatti Antropici e Sostenibilità in ambiente marino (IAS) of CNR, 09170 Oristano, Italy,
https://orcid.org/0000-0002-6709-1600, https://orcid.org/0000-0003-0268-7822
[2]Istituto di Scienze Marine (ISMAR) of CNR, 19032 La Spezia, Italy, https://orcid.org/0000-0002-5654-4731

*Correspondence to*: Alberto Ribotti (alberto.ribotti@cnr.it)

**Abstract.** Seven oceanographic cruises in five years were organized in the Sardinia Sea with the repeated collection of physical, chemical and biological data. An accurate and sustained quality assurance on physical sensors was acted through prior and post-cruise calibration and verified during in-situ acquisitions with the use of redundant sensors and other instruments. Moreover, for dissolved oxygen and conductivity, seawater samples at standard depths were frequently analyzed on-board. Then an accurate quality control was used to verify all hydrological data profiles, that passed a further quality check following standard procedures. Finally all hydrological data have been included in two online public open access datasets in the SEANOE repository (https://doi.org/10.17882/59867 and https://doi.org/10.17882/70340, Ribotti et al., 2019a,b). During and after all cruises also chlorophyll and nutrients analyses were carried on but data are not yet open access; the same for water current profiles, both at casts and during vessel moves, and geophysical data. These ocean data are the first covering the whole Sardinia Sea for its whole extension. Here data and assurance/control procedures used are described as they became standards in deep sea acquisitions in the years.

**Keywords:** Sardinia, CTD, hydrological data, data quality check, sensor calibration

## 1 Introduction

Between May 2000 and January 2004 the National Research Council (CNR) of Italy collected hydrodynamic, chemical and biological data in the Sardinia Sea during seven multidisciplinary oceanographic cruises, named from medgoos1 to medgoos7. These cruises were the first covering the whole Sardinia Sea, from the shelf to the open sea, with oceanographic measurements. They were realized with the main aim to give a useful contribution on the knowledge of the local upper, intermediate and deep circulation and its interaction with the general Mediterranean circulation.

The study area is limited between 38 and 42 °N in latitude and between 7 °E and the western Sardinian coast in longitude, with an offshore bottom reaching a depth of 2950 m in the abyssal plain. The shelf extends from 5.5 km at north to 37 km in its centre with a shelf break at about 200 m depth (Conforti et al., 2016; Brambilla et al., 2019). It is an area of passage of re-



circulating waters between the two Mediterranean sub-basins and the Atlantic Ocean (Astraldi et al., 1999; Millot, 2005;
Millot and Taupier-Letage, 2005; Schroeder et al., 2013) and is strongly influenced by the Algerian large scale dynamics
(Bouzinac and Millot, 1999; Puillat et al., 2002; Pessini et al., 2018). Very energetic surface fronts, cyclonic and anticyclonic
vortices, up and downwelling events play an important role in variability and transport of physical, biological and chemical
characteristics of the water masses (Puillat et al., 2003, 2006; Santinelli et al. ,2008; Olita et al., 2013, 2014). Here, thanks to
the data acquired during the first medgoos cruises, Puillat et al. (2003) and Ribotti et al. (2004) identified the main water
masses usually retrieved in the rest of the western Mediterranean sea, like the Atlantic Water (AW) in the upper 150 m, the
Winter Intermediate Water and its modified version known as the Temperature Minimum Layer (Benzohra and Millot, 1995;
Sorgente et al., 2003; Allen et al., 2008) at about 100-120 m depth in the AW, the Intermediate Waters below to 800 m depth
and the Western Mediterranean Deep Waters (WMDW) to the bottom. Here, since 2005 the old WMDW has been
undermined by the warmest and saltier new WMDWs that diffused all over the Western Mediterranean sub-basin due to the
transfer of the Eastern Mediterranean Transient signal inside. This water is characterized by high heat and salt contents from
the advected LIW, originating the new WMDW in the north-western Mediterranean sub-basin (Gasparini et al., 2005;
Schroeder et al., 2006; Zunino et al., 2012; Ribotti et al., 2016).

Despite of such an importance and apart the above mentioned medgoos cruises, just a few in-situ measurements were
organized and ocean data acquired in this area in the years, apart French or Italian unsystematic cruises since the '50s to
nowadays and available in the PANGAEA (https://www.pangaea.de) and SEANOE repositories (Dumas et al., 2018). In
2014, NATO STO-CMRE based in La Spezia organized a 2-week experiment with a large use of ocean instruments and two
research vessels in a limited area (110x110 km$^2$). Its aim was to improve local ocean numerical simulations and forecasts and
study the local ocean variability and structures (Onken, 2017a,b; Knoll et al., 2017; Hemming et al., 2017; Onken et al.,
2018; Hernandez-Lasheras and Mourre, 2018).
Then recent experiments with drifters and deep sea gliders were realized in the Sardinia Sea and between Sardinia and
Balears, partially described by Olita et al. (2014).

In 2000-2004, instruments, sensors and data passed severe controls following internationally accepted oceanographic
processes, necessary to obtain high-quality data. They changed and were adapted depending by several factors like the
working environment, the oceanographic instruments/sensors used, the type of data acquired. These practices included
quality assurance, control and assessment, standards and best practices refined till nowadays (Hood et al., 2010; Bushnell et
al., 2019; Pearlman et al., 2019). In this paper we describe all the procedures or best practices followed to assure and control
the quality of the acquired data during the medgoos cruises, the sensors used, their calibrations and intercomparisons.

Acquired hydrological data are in two datasets stored in an open access repository, called SEA scieNtific Open data Edition
(SEANOE) (Ribotti et al., 2019a,b), linked with the EMODnet Data Network of marine centers and European thematic data
portals like SeaDataNet and EurOBIS.

In the following two paragraphs (2 and 3) vessel, instruments and sensors used in the seven cruises are described, with a
distinction between data in repositories (par. 2) and not part of any repository (par. 3). The calibration of temperature and



conductivity sensors are part of the paragraph 4 while in paragraph 5 the on board control of CTD sensors stability is detailed. Discussion and conclusions close the paper.

## 2 Instrumentation technology

During the seven cruises profiles of physical/chemical parameters were acquired at planned stations whose number varied due to the length and the strategy adopted at each cruise, the wideness of the covered area, and the sea conditions. So the activities range from the 38 stations realized during the 6-days-long medgoos1 in 2000 and the 92 during the 21-days-long medgoos6 in 2003. In September 2001, just the southern part of the Sardinia Sea with 41 stations was covered due to bad weather conditions.

The 61,30 meters long R/V Urania of CNR was used in all cruises. This was a modern multidisciplinary research vessel equipped with instruments to study physical and chemical water quality parameters, and laboratories for biological and geological analyses. For its dynamic positioning the vessel was equipped with an integrated navigation system constituted by two DGPS antennas and one Loran C that ensured an optimal use of the scientific equipment during the cruises. Such a system was managed through a software by Andrews Hydrographics installed on PC HP386, 33 MHz that permitted to download navigation and meteorological data in ASCII format with geographic and kilometric coordinates at frequency till 10 minutes.

On board, a SBE911 plus CTD probe (by Sea-Bird Inc.), mounted on a 24 10-liters Niskin bottles rosette for water column sampling, was used to acquire hydrological data during all the seven cruises (figure 1).

In specific, the sensors installed on the probe had the following characteristics (table 1):

• pressure [db]: a Digiquarz 4000 pressure transducer was used. The transducer had a resolution of 0.01 ppm, oscillator frequency 34 KHz - 38KHz and temperature range 0 °C - 125 °C;

• water temperature [deg C]: a SBE-3/F thermometer with response time of 70 ms, temperature range -5 ° - + 35 °, accuracy about 0.004 ° C per year, resolution 0.0003 ° C. The international practical temperature scale known as IPTS-68 was applied on data from medgoos1, 2 and 6 while the international temperature scale of 1990, known as ITS-90, on data from medgoos4 and from medgoos3, 5, 7 on secondary sensors;

• conductivity [mS/cm]: a SBE-4 sensor with a range of 0.0 - 7 S/m, resolution 0.00004 S/m, accuracy about 0.0003 S/m per month and response time of 0.085 sec with pump or 0.17 sec without pump;

• dissolved oxygen: a SBE-13 Beckman/YSI sensor with a range of 0 - 15 ml/l, accuracy of 0.1 ml/l, resolution 0.01 ml/l and response time 2 sec at a temperature of 25 °C during cruises medgoos1, 2, 3, 4. During cruises



medgoos5, 6, 7 a SBE-43 polarographic membrane sensor for pumped CTD applications with titanium (7000 m) housing was used with a range of 120% of surface saturation in all natural waters, accuracy of ±2% of saturation;

• fluorescence: a Sea Tech Inc. fluorometer with energy emitted by the flash lamp of 0.25 J for flash, temperature range 0 - 25 °C, resolution 0.15 μg/l.

Redundant or secondary sensors were always used for a data quality assessment (as defined in Bushnell et al., 2019) of both temperature and salinity measurements, apart during the cruise medgoos1 in May-June 2000 (see table 1). A secondary SBE-43 sensor for dissolved oxygen was also added just in the last three cruises, from medgoos5 to 7. These redundant sensors were a useful method of comparison to evaluate the stability of primaries both during the acquisition and at a following visual quality check of profiles.

For the same reason, digital deep sea reversing thermometers RTM 4002 by Sensoren Instrumente Systeme GmbH (SiS) were mounted on Niskin bottles during medgoos4 (3 thermometers) and 5 (4 thermometers). These instruments acquired sea temperature at depths, defined by the closing of the Niskin bottle where they were mounted on, usually sampling near the bottom where temperature is more stable. Reversing thermometers had a depth range of up to 10000 meters and a temperature range between -2 and +40 °C. They had a resolution of 105 ±0.001 °C between -2.000°C and 19.999 °C, of ±0.01 °C between 20.00 °C and 40.00 °C and a stability of 0.00025 °C per month. Its pressure housing was made of a glass tube closed at its ends by metal stoppers, one containing the platinum sensor and the other the battery. Its internal mercury switch was activated by inverting the instrument at defined depths.

**3 Other acquisitions**

In some cruises (see table 1) currentmeters data were acquired on both at a station (Lowered Acoustic Doppler Current Profiler or LADCP) and in route (Shipborne Acoustic Data Current Profiler or SADCP) while geophysical sub-bottom profiler data just during the cruise medgoos2. At casts, the real time display of CTD data made it possible to identify a certain number of stations where to take water samples for the estimation of nutrients (nitrites, nitrates and phosphates), Chlorophyll-α, chromophoric dissolved organic matter (CDOM) and 115 dissolved organic carbon (DOC). As all these data are not in the two datasets in the SEANOE repository mentioned above (Ribotti et al., 2019a,b), they will be shortly described here as part of the amount of cruises data. All measures were carried out trying to reconcile the different procedures.





The Sub-Bottom Profiler was a GeoPulse Transmitter Model 5430A at frequencies ranging between 2 KHz and 15 KHz with a maximum emitted power of 10 KW. It was used on a small portion of the western Sardinian shelf

north of Oristano just in April 2001 during the cruise medgoos2.

Starting from the cruise medgoos2, profiles of current speed were acquired during CTD casts by two synchronized 300 kHz RDI Workhorse ADCPs, by RD Instruments Inc. in USA (now Teledyne), configured in modality Lowered and installed on the rosette one looking up (named slave) and the second down (named master). They acquired horizontal current data in 20 cells 10 m width each from the instrument with an impulse

per second. Then under the keel of the vessel, a 38 kHz ADCP profiled currents in 8 m wide cells over 1000 m depth through an impulse per second during transfers between stations. Another impulse was used to correct the water speed and obtain its real speed as regards to the sea bottom (bottom tracking). Configuration of ADCPs used in the two modes, Lowered and Shipborne, in 2001-2004 are in agreement with the more recent internationally recognized GO-SHIP protocols described by Hood et al. (2010).

For nutrients, Chlorophyll α, CDOM and DOC all the water samples were filtered to remove the particulate fraction immediately after collection and frozen at different temperatures (+4 ° C for DOC samples and -20 ° C for nutrients) to be analyzed at labs following standard procedures like that described in Strickland and Parsons (1972) for nutrients, in Lazzara et al. (1990) for Chlorophyll-α, in Vignudelli et al. (2004) and Santinelli et al. (2008) for CDOM and DOC. In the first three medgoos cruises nutrient samples were partially analyzed on-board

by a Systea μCHEM Auto-analyzer.

## 4 Pre- and post cruise calibration procedures

The pre- and a post-cruise calibration of the sensors of temperature and conductivity was performed at the oceanographic instrument (CTD) calibration facilities of the SACLANT Undersea Research Center (SACLANTCEN, now STO-CMRE) in La Spezia, Italy. The Centre was funded in 1959 initially for submarine

warfare but it developed and maintained unique, in Italy and for years, an oceanographic instruments test and calibration facility that enabled the acquisition of high-quality ocean data. Two calibration seawater tanks were equipped with two very high precision Neil Brown ATB-1250 Platinum Resistance Thermometer Bridge (figure 2) for temperature and two very high precision Neil Brown CSA-1250 conductivity/salinity adaptor for conductivity. Seawater samples were analyzed for conductivity by highly précised 8400B Autosal Laboratory

Salinometer, from Guildline Instruments Ltd™, standardized with IAPSO Standard Seawater and an accuracy of <0.002 psu on a range of salinity between 2 and 42 psu.



Before SBE sensors calibration, the two temperature sensors in the bath were adjusted to a triple-point-of-water cell (TPW) at the temperature of 0.01 °C and a thermometric standard Gallium-melting-point cell for the highest value of 29.7646 °C. So at SACLANTCEN a calibration of these sensors was realized exceeding WOCE

standards (Millard and Yang, 1993). These two calibrations permitted to substantially generate a slope correction, used in the configuration file of the SBE Seasoft™ suite of programs, for data acquired during each cruise then improving their quality.

## 5 On board control of CTD sensors stability

Despite the calibration of temperature and conductivity sensors before and after each cruise, all sensors can significantly drift

over the course of a cruise. This can dramatically reduce the quality of the data. As high quality conductivity and oxygen data permit to define local water masses with high precision, particularly in the deep, then the use of international standards is mandatory in oceanography. The stability of conductivity and dissolved oxygen sensors must be verified on-board through the comparison with data from water samples. During the cruise all involved personnel and all acquired data are usually together so it is easier to check and correct repetitive problems before they can further degrade the data.

Conductivity data were checked against the on-board analyses by a Guildline™ 8400B Autosal Laboratory Salinometer, similar to that described above, while dissolved oxygen data against Winkler titration method with a measured precision in triplicate analyzed sample of 0.01ml/l (expressed in standard deviation). The sampling was at defined depths at surface, 25, 50, 100, 200, 300, 400, 500, 750, 1000, 1250, 1500, 1750, 2000, 2500, 3000 metres, and bottom.

The Autosal salinometer was operated in a small temperature-regulated room part of the on board wet laboratory and its bath

temperature was held at 24±2 °C (figure 3 left). The salinometer was daily standardized with the use of IAPSO Standard Seawater provided by 200 ml clear sealed glass bottles, before starting the analyses. Seawater for conductivity was taken after that for dissolved oxygen and collected in 250 ml clear bottles with screw cap. Each bottle and its cap were rinsed three times with the sampled water, and filled to its shoulder. Then they were stored longer than 24 hours in the conditioned room before their analyses. Niskin bottles collected water at the deepest casts of the cruise and at least once a day.

The oxygen sensors used during the cruises were not calibrated, so during each cruise different verifications were realized (figure 3 right) to verify possible sensors shifts through the Winkler titration method (Winkler, 1888). This method, used for in-situ dissolved oxygen analyses, consists of reacting oxygen in the samples with two reagents (I and II) and with a final titration. The reagent I is a solution of $MnSO_4$ and NaOH while the reagent II is a solution of NaI e $H_2SO_4$. The utmost attention is paid to draw the oxygen samples first from Niskins and into dark glass bottles. This avoids the formation of air

bubbles during the sampling itself or the execution of the analysis. The sampling was realized as for conductivity for frequency and methodology. The water sampled at 100 m of depth was used to obtain three blank solutions in dark glass bottles and one standard solution in a larger plastic bottle. So blanks and standards were run often. Then different quantities





of reagents were added: 0.5 ml (the usual for all samples) of reagent I and reagent II in the first blank, its double in the second blank and the triple in the third. No reagents were initially added in the standard solution. All samples and the three

blank solutions, not the standard, were left in the darkness for at least 2 hours and then analysed within 24 hours from their collection. During the analysis of the standard solution, reagents I and II, 0.5 ml of H2SO4 and then of KIO3 were added. The samples have been analysed through the programme TIAMO 2.0™, that stands for *TItration And MOre*», by Metrohm. Thanks to the data from the water samples, the possible estimations of changes of slope (Soc) and offset (Voffset), in the linear relationship between oxygen concentration and voltage output that indicate a loss of sensitivity of the sensors, were

realized performing a linear regression line of the data calculated from the following Eq. (1) (Owens and Millard Jr, 1985; Millard and Yang, 1993; SBE, 2008):

$$\varphi = Oxsol(T,S) * (1.0 + A * T + B * T^2 + C * T^3) * e^{\left(\frac{E*P}{K}\right)} \qquad (1)$$

where Oxsol(T,S) is the oxygen solubility (ml/l) at a defined temperature and salinity, T and K are the CTD temperatures in °C and in °K, P is the CTD pressure (dbars) and A, B, C, E are calibration coefficients.

At a defined deep station, the regression is calculated by using the measured Winkler oxygen concentration divided by φ as dependent variable and the oxygen sensor output voltage as independent variable.

In figure 4 an example of linear regression applied on a deep CTD station with the calculation of the new Soc (0,3936) and the new Voffset = -0,2094/SOC = -0,2094/0,3936 = -0,53201

During cruises from medgoos4 to medgoos7 temperature data were checked at defined depths against reverse thermometers

(figure 5) installed in correspondence of the bottles number 1, 3, 5, 7 of the rosette sampler.

Furthermore CTD and oxygen data were compared with those analyzed on board from samples or acquired with other instruments in order to determine or visually check possible shifts. Acquired CTD data were processed through the standardized procedures of the SBE Data Processing™ software. After comparisons and in case of malfunction, the use of the secondary sensor instead of the primary was evaluated (in figure 6 a plot for conductivity and dissolved oxygen).

After the post-cruise calibration, if the shift was constant or systematic then an average of all data (primary, secondary, samples, etc) was used to correct data. If shift was random, a trend was considered for data correction.

Finally Chlorophyll-a fluorescence (Chl-α) and pressure sensors were not calibrated. The first is reported as Relative Fluorescence Unit (R.F.U.) in the datasets. The pressure sensor is usually stable but if problems occurred it was sent to SBE Inc. in USA for its calibration.

**6 Data availability**

The two datasets described in this study are publicly available and free of charge from the SEANOE data repository (Ribotti et al., 2019a, https://doi.org/10.17882/59867 and Ribotti et al., 2019b, https://doi.org/10.17882/70340). The presented datasets are composed of CTD data in Ocean Data View (ODV) TXT Spreadsheet and Collection files formats divided per cruise. Metadata are available in TXT, RIS, XLS, RTF, BIBTEX formats. Data and metadata from all cruises, apart



medgoos1, are also stored and available under request in the Mediterranean Marine Data at http://www.mediterranean-marinedata.eu/, a CNR - ENEA collaborative initiative with the aim to archive and distribute oceanographic data and information.

## 7 Discussion and conclusion

Several processes to obtain high-quality ocean data were followed during the 2000-2004 medgoos cruises in the Sardinia
Sea, western Mediterranean. Quality assurance, control, assessment, standard and best practices, defined at international level (see Bushnell et al., 2019) and after decades of practices in ocean data acquisition, were considered during all cruises and are in agreement with recent standardized procedures for all sensors (Hood et al., 2010). Sensors pre and post-calibration, use of redundant ones, comparisons with on-board analyzed water samples showed their efforts resulted in achieving the required accuracy standard for that period, due to sensors accuracies, and for today. Uncertainty of
measurements, defined as "the quantification of the doubt that exists about results of any measurements", exists in these data but reduced by the skill of the same operators or analysts following all the processes at any time for all the years and cruises. Now oceanographic data from seven cruises are collected in two open access datasets available online, including ocean parameters like conductivity, temperature, dissolved oxygen and Chlorophyll-α fluorescence. These ocean data are the first available for the Sardinia Sea, important to characterize the general circulation and the dynamics in the western
Mediterranean.

## Author contribution

AR led some cruises, organized the two datasets and led the writing of the paper. MB led some cruises, finalized QA and QC procedures described in the paper and collaborated to the paper writing. RS collaborated to the paper writing.

## Competing interests

The authors declare that they have no conflict of interests.

## Acknowledgments.

The data used in this work have been collected in the framework of national and European projects like the Italian MIUR project SIMBIOS (Operative Programme of the Marine Environment Plan, Cluster C10, Project n. 13 – D.n. 778.RIC); the EU Marie Curie Host Fellowship ODASS (HPMD-CT-2001-00075); the EU MAMA (EVR1-CT-2001-20010), the EU
MFSPP (MAS3-CT98–0171); the EU MFSTEP (EVK3-CT-2002-00075). Authors thank the chiefs and the crews onboard the R/V Urania during the mentioned cruises for essential their support, Dr. M. Di Bitetto as CNR cruises leader/co-leader and





Dr. S. Vallerga and Dr. R. Sorgente as CNR in charge of funding projects. We want also to thank Dr. G.M.R Manzella from ENEA in La Spezia (Italy) to have created and maintained the Mediterranean Marine Database for years.

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

**Figure Captions**

**Figure 1. All the CTD casts during the seven cruises in the Sardinia Sea (western Mediterranean)**

**Figure 2. An ATB-1250 Platinum Resistance Thermometer Bridge. [FOTO]**

**Figure 3. The on-board laboratory for conductivity (left) and dissolved oxygen (right) analysis from seawater samples during the seven cruises.**

**Figure 4. Linear regression line calculated for the dissolved oxygen data of a CTD station**

**Figure 5. A reverse thermometer on a Niskin bottle of the rosette**

**Figure 6. The salinity (up) and dissolved oxygen (down) errors between what measured by the sensor on the probe and what measured on board from water samples for the two conductivity/oxygen sensors, 1 and 2, along a vertical profile in a deep station.**

**Table Captions**

**Table 1. The list of sensors used and activities realized during the seven medgoos cruises. Numbers in brackets for reversed thermometers are the numbers of the Niskin bottles where installed. L and SADCP stand for Lowered and Shipborn Acoustic Current Profile, respectively**





**Figure 1**

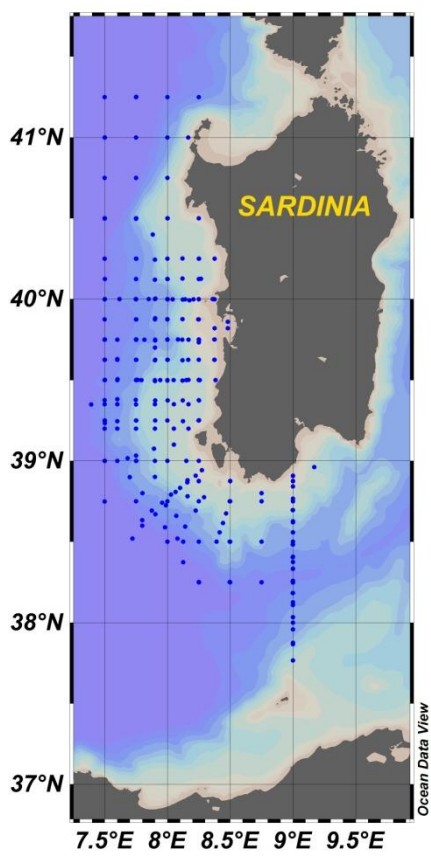


**Figure 2**

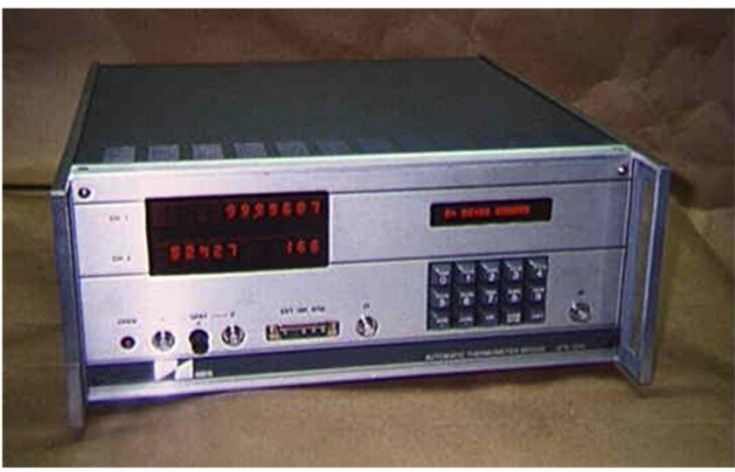



**Figure3**

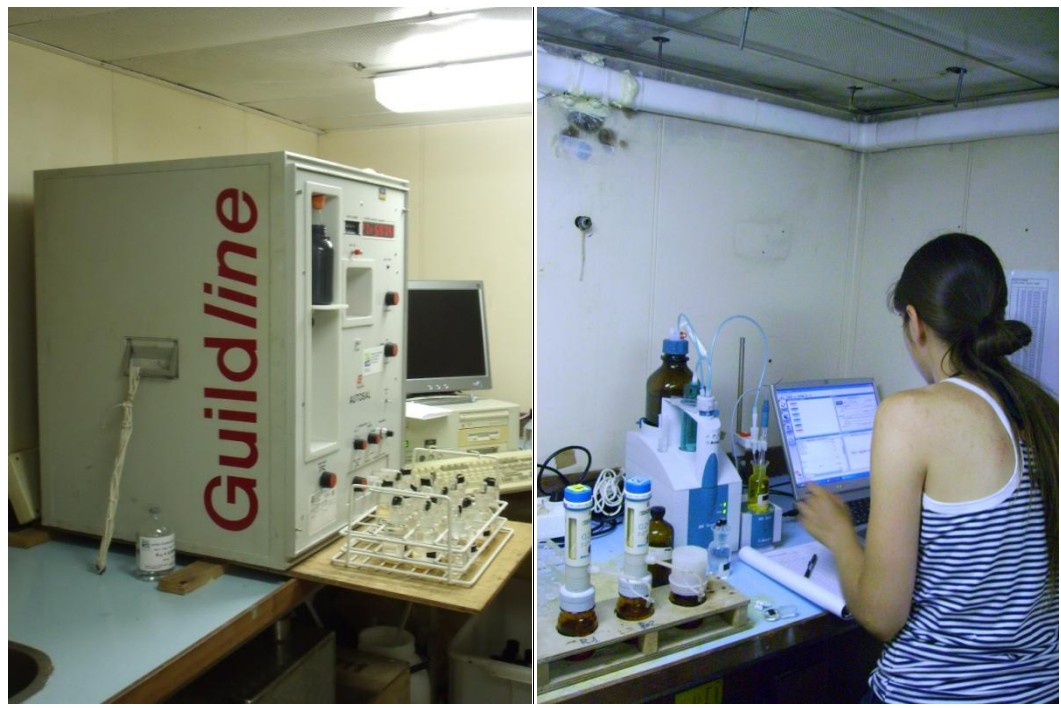


**Figure4**

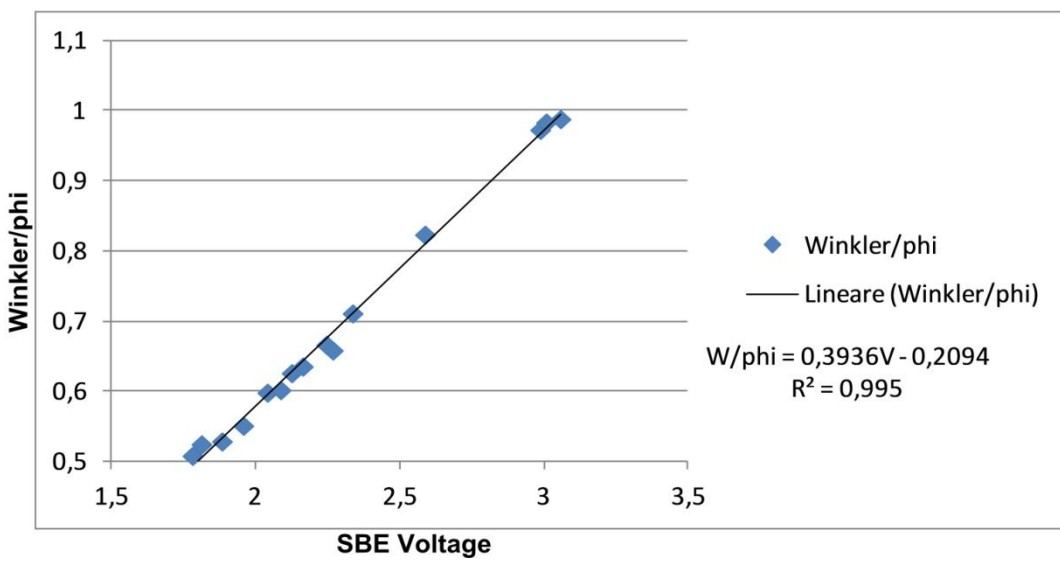





**Figure5**

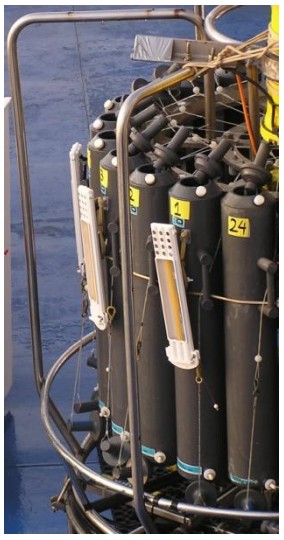

**Figure6**

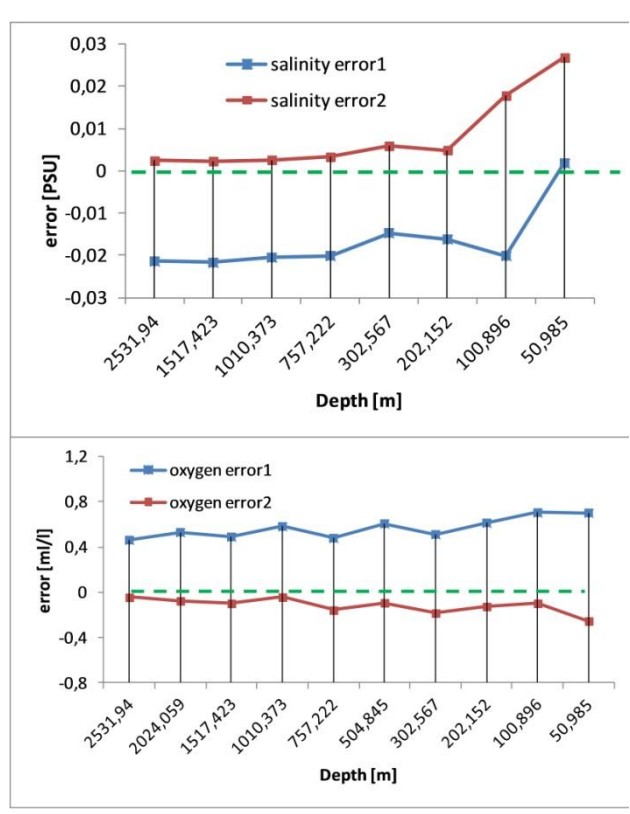

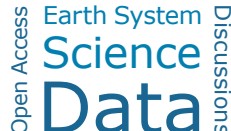

**Table 1**

| | medgoos1 | medgoos2 | medgoos3 | medgoos4 | medgoos5 | medgoos6 | medgoos7 |
|---|---|---|---|---|---|---|---|
| **Date** | 28/05-02/06/2000 | 23/03-03/04/2001 | 10-20/09/2001 | 4-23/05/2002 | 31/10-18/11/2002 | 28/03-17/04/2003 | 07-26/01/2004 |
| **# days** | 6 | 12 | 11 | 20 | 20 | 21 | 20 |
| **R/V** | Urania | Urania | Urania | Urania | Urania | Urania | Urania |
| **# CTDs** | 38 | 67 | 41 | 68 | 42 | 92 | 87 |
| **Reverse Thermometer (bottle #)** | | | | 3 (1, 3, 5) | 4 (1, 3, 5, 7) | | |
| **SBE-13 O2 Sensor** | X | X | X | X | | | |
| **SBE-43 O2 Sensor** | | | | | X | X | X |
| **Fluorescence** | X | X | X | X | X | X | X |
| **LADCP** | X | X | X | X | X | | |
| **SADCP** | X | | | | | | |
| **CTD secondary sensors** | | | | | | | |
| **Secondary Temperature** | | X | X | X | X | X | X |
| **Secondary Conductivity** | | X | X | X | X | X | X |
| **SBE-43 O2** | | | | | X | X | X |
| **Water samples for following analyses of** | | | | | | | |
| **DOC** | 23 | 66 | 39 | | 55 | | |
| **Nutrients** | 23 | 66 | 39 | | 55 | | |
| **Phytoplancton** | | X | X | X | X | | |
| **Chl-α** | | 66 | 39 | X | 55 | | |
| **On-board analyses** | | | | | | | |
| **O2** | X | | X | X | X | X | X |
| **Conductivity** | X | | X | X | X | X | X |
