# Peer review of "Quality assurance and control on hydrological data off western Sardinia (2000 - 2004), western Mediterranean"

_Earth System Science Data, 2020_

## Referee Comment (RC1) · Alain Lefebvre (Referee) · 12 Feb 2020

Dear colleague, please find attached 2 files : my reviewer comments + Alberto Ribotti's manuscript with annotations Best regard, AL

Please also note the supplement to this comment:
https://www.earth-syst-sci-data-discuss.net/essd-2020-17/essd-2020-17-RC1-supplement.zip

---

## Referee Comment (RC2) · Sarantis Sofianos (Referee) · 10 Mar 2020

The manuscript "Quality assurance and control on hydrological data off western Sardinia (2000 - 2004), western Mediterranean" by Ribotti et al. presents an open access dataset (and its quality control methods) that includes data acquired during seven oceanographic cruises off the coast of western Sardinia, Western Mediterranean. The main aim of the cruises and the publication of the dataset is to give to the scientific community a contribution to the local circulation and interaction with the general Mediterranean circulation. The description of instrumentation and quality assurance and control methods of the dataset is clear and informative (although the pictures of

specific instruments seem redundant). The information is also included in the SEA-NOE archives. Based on the aim of the cruises, the absence of hydrodynamic (ADCP) data makes the dataset "weaker" and less informative. The reason for not including this data should be better explained by the authors.

Overall, the manuscript follows the scope and recommendations of the journal and the dataset can be useful for future research and assessment work of the oceanographic community. Thus, I propose that the paper should be accepted after minor revisions.

Minor comments:

Line 15: "carried out" (instead of "carried on") Line 28: "center" (instead of "centre") Line 35: "observed" (instead of "retrieved") Line 44: apart from Lines 83-84: units are not correct Line 96: "except" (instead of "apart") Line 156: "enable the definition of" (instead of "permit to define")

---

## Author Comment (AC1) · 21 Apr 2020

Dear Dr. LEFEBVRE, we thank the you for having accepted the review and improved our manuscript with your comments and suggestions. We agree with the points raised and we have answered all your comments.

Below, our responses (A:) follow your comments (R:).

R: Cruises have been defined to "give a useful contribution on the knowledge of the local upper, intermediate and deep circulation and its interaction with the general Mediterranean circulation". Considering these objectives, what is surprising is that

[Figure]

proposed variables do not include systematic current measurements but main classical physical and biological variables. Authors should explain why such current measurements were not implemented systematically. Indeed, in section 3. Other acquisition, we understand that current meters were deployed only during a limited number of cruises.

A: Thanks for having raised this point. LADCP measurements at CNR started in those years and ADCPs were not available during all cruises. After their acquisition, usually current data were stored but never published. A great job is ongoing in these months to rescue and reprocess all raw data and build a complete high quality database. The following sentence has been added in the 6th paragraph (p.7 lines 211-213): "The rescue and reprocessing of hydrodynamic data is ongoing in order to reorganize them in a quality controlled database. The same will be for the biological parameters. A first dataset of nutrients data from more recent cruises (late 2014 - 2017) is available at https://doi.org/10.5194/essd-2019-136."

R: A link with the Essential Oceanographic Variables (EOV) and Essential Biodiversity Variables (EBV) as recommended for a well-suited monitoring programme should have been interesting to develop.

A: We fully agree with the reviewer and this was an error.

R: The materials and methods are described in sufficient details. The authors provided main characteristics of each sensor and also the associated range, accuracy, resolution and response time. The quality of the data is assured as the authors follow (well documented) good practises at sea and in lab. All these information are also included in the two associated SEANOE archives. We can however regret the absence of a validation protocol for fluorescence data.

A: Fluorescence sensor was calibrated following the instructions described by Sea-Bird Electronics Inc. in its APPLICATION NOTE NO. 9, downloadable at https://www.seabird.com/application-notes, where SEASOFT Coefficients for the Sea Tech Fluorometer are calculated. However no validation of the data has been applied

after the cruises. Anyway the sentence was rewritten as follows, adding a new reference (see p.7 new lines 199-202): "Chl-a was calibrated following the instructions given by SBE (2008a) where the SEASOFT^{TM} coefficients for the Sea Tech fluorometer are calculated. No validation of the data has been applied after the cruises. It is reported as Relative Fluorescence Unit (R.F.U.) in the datasets. Also pressure sensors were not calibrated as usually stable but, if problems occurred, it was sent to SBE Inc. in USA for its calibration." Added the following reference in bibliography "SBE: AN-9: Calculating SEASOFT Coefficients for Sea Tech Fluorometer and WET Labs Flash Lamp Fluorometer (FLF). 1 p, 2008a"

R: The data set format as proposed in SEANOE is appropriate for immediate use. I'm not sure that figures 2, 3 and 5 are very helpful.

A: We thought that figures could improve the description of the instruments or labs on the text but, as both reviewers noted a redundancy in description plus figures, we have deleted them. Then the numbers of the figures have been updated in the text.

R: To conclude, I propose that the manuscript should be accepted subject to minor revisions.

A: Minor corrections and comments directly in the text as comments in the attached PDF file: New pages and lines are mentioned if they changed after correction as new p.x and7or new line y. p.2, line 43: put in-situ in italic in the whole text. Here it disappeared due to the rewriting of the whole sentence.

p.2, line 44, now lines 43-44: whole sentence rewritten accordingly "Due to such an oceanographic importance, since the '50s French or Italian unsystematic cruises have been organized in the area whose data are available in the PANGAEA (https://www.pangaea.de) and SEANOE repositories (Dumas et al., 2018)."

p.2, line 46, now lines 45-46: acronym defined in the text. Sentence rewritten as "the Science and Technology Organisation - Centre for Maritime Research and Experimen-

tation of the North Atlantic Treatment Organization (NATO STO-CMRE), based in La Spezia,"

p.2, line 53: sentence deleted as it did not improve the information on instruments or data then previously

p.3, line 69, now line 68: changed accordingly

p. 3, line 71, now line 70: I would leave the past "was" as this vessel does not exist anymore

p.3, line 83: changed in the whole text for all parameters

p. 5, line 130, now line 126: changed accordingly

p.5, line 130, now lines 126-132: added what requested but rewriting the sentences as follows "For Chlorophyll-a (Chl-a), CDOM and DOC all the water samples were filtered on 0.42$\mu$m Whatmann GF/F glass microfiber filters for Chl-a and 0.22$\mu$m membrane filter (Sartorius, Minisart, SM 16534 K) for DOC and CDOM; this to remove the particulate fraction immediately after collection. Then filters were frozen at +4°C to be analyzed at labs following standard procedures described by Lazzara et al. (1990) for Chl-a, and by Vignudelli et al. (2004) and Santinelli et al. (2008) for CDOM and DOC. For nutrients, during the first three medgoos cruises water samples were partially analyzed on-board by a Systea ïĄ■CHEM Auto-analyzer, while remaining samples frozen at -20°C following Strickland and Parsons (1972). In the following cruises all water samples were frozen and then analyzed once back."

p.5, line 135, now line 132: checked in the whole text and reformatted following the Copernicus Word Template

p. 6, line 168, now line 164: corrected

p.6, line 173, now line 168: corrected

p.6 line 175, now 170: corrected

p.7 line 180, now p.6 line 175: corrected

p.7 line 181, now p.6 line 176: corrected

p.7 line 201, now p.6 line 196: rewritten as "If the shift was random, Soc and Voffset for dissolved oxygen, slope and offset for conductivity/temperature were recalculated and then the data corrected following the procedures described in the application notes number 64-2 (SBE, 2008b, 2012) and 31 (SBE, 2016), respectively." Added the following references in bibliography " SBE: AN-64-2: AN64-2: SBE 43 Dissolved Oxygen Sensor Calibration and Data Corrections. 5 pp, 2012" and "SBE: AN-31: Computing Temperature & Conductivity Slope & Offset Correction Coefficients from Lab Calibration & Salinity Bottle Samples. 8 pp, 2016"

p.12 line 355: figure deleted

p.12 lines 356-357: figure deleted

p.12 line 359: figure deleted

p. 14 ex-figure 4, now figure 2: changed decimal separator accordingly

p. 15 ex-figure 6, now figure 3: changed decimal separator and depth format accordingly

p. 16 table 1: changed decimal data format according to ISO 8601

Please also note the supplement to this comment:
https://www.earth-syst-sci-data-discuss.net/essd-2020-17/essd-2020-17-AC1-supplement.pdf

[Figure]

SARDINIA

*Ocean Data View*

**Fig. 1.**

[Figure]

[Figure]

**Fig. 2.**

[Figure]

**Fig. 3.**

---

## Author Comment (AC2) · 21 Apr 2020

Reviewer's 2 comments on the following manuscript: essd-2020-17, Submitted on 10 March 2020 Quality assurance and control on hydrological data off western Sardinia (2000–2004), western Mediterranean. by Alberto Ribotti, Roberto Sorgente, and Mireno Borghini Reviewer: Dr. Sarantis Sofianos (Referee)

Dear Dr. SOFIANOS, we thank the you for having accepted the review and improved our manuscript with your comments and suggestions. We agree with the points raised and we have answered all your comments.

Below, our responses (A:) follow your comments (R:).

R: The manuscript "Quality assurance and control on hydrological data off western Sardinia(2000 - 2004), western Mediterranean" by Ribotti et al. presents an open access dataset (and its quality control methods) that includes data acquired during seven oceanographic cruises off the coast of western Sardinia, Western Mediterranean. The main aim of the cruises and the publication of the dataset is to give to the scientific community a contribution to the local circulation and interaction with the general Mediterranean circulation. The description of instrumentation and quality assurance and control methods of the dataset is clear and informative (although the pictures of specific instruments seem redundant).

A: We thought that figures could improve the description of the instruments or labs on the text but, as both reviewers noted a redundancy in description plus figures, we have deleted them. Then the numbers of the figures have been updated in the text.

R: The information is also included in the SEANOE archives. Based on the aim of the cruises, the absence of hydrodynamic (ADCP) data makes the dataset "weaker" and less informative. The reason for not including this data should be better explained by the authors.

A: Thanks for having raised this point. As explained to referee 1 too, LADCP measurements at CNR started in those years and ADCPs were not available during all cruises. After their acquisition, usually current data were stored but never used. A great job to rescue, reprocess all raw data and build a complete high quality database is ongoing in these months.

The following sentence has been added in the 6th paragraph (p.7 lines 211): "The rescue and reprocessing of hydrodynamic data is ongoing in order to reorganize them in a quality controlled database."

Minor corrections:

p.1 line 15: corrected

p.1 line 28: corrected

p.2 line 35: corrected

p.2 line 44: apart from the sentence was completely rewritten

p. 3 lines 83-84, now lines 82-83: units corrected

p.4 line 96, now line 95: corrected

p.6 line 156, now line 151: corrected

Please also note the supplement to this comment:
https://www.earth-syst-sci-data-discuss.net/essd-2020-17/essd-2020-17-AC2-supplement.pdf

SARDINIA

*Ocean Data View*

7.5°E  8°E  8.5°E  9°E  9.5°E

41°N

40°N

39°N

38°N

37°N

**Fig. 1.**

[Figure]

[Figure]

**Fig. 2.**

**Fig. 3.**

---

## Author Response (AR1)

**Reviewer's 1 comments on the following manuscript:**

essd-2020-17,  Submitted on 24 Jan 2020
**Quality assurance and control on hydrological data off western Sardinia (2000–2004), western Mediterranean.**
**by Alberto Ribotti, Roberto Sorgente, and Mireno Borghini**

**Reviewer**: Dr. Alain LEFEBVRE, Ifremer, France (Alain.Lefebvre@ifremer.fr)

Dear Dr. LEFEBVRE,

we thank the you for having accepted the review and improved our manuscript with your comments and suggestions.

We agree with the points raised and we have answered all your comments.

Below, reviewer's comments are given in normal font and our responses in red.

Cruises have been defined to "give a useful contribution on the knowledge of the local upper, intermediate and deep circulation and its interaction with the general Mediterranean circulation". Considering these objectives, what is surprising is that proposed variables do not include systematic current measurements but main classical physical and biological variables. Authors should explain why such current measurements were not implemented systematically. Indeed, in section 3. Other acquisition, we understand that current meters were deployed only during a limited number of cruises.

Thanks for having raised this point. LADCP measurements at CNR started in those years and ADCPs were not available during all cruises. After their acquisition, usually current data were stored but never published. A great job is ongoing in these months to rescue and reprocess all raw data and build a complete high quality database.

The following sentence has been added in the 6th paragraph (p.7 lines 211-213): "The rescue and reprocessing of hydrodynamic data is ongoing in order to reorganize them in a quality controlled database. The same will be for the biological parameters. A first dataset of nutrients data from more recent cruises (late 2014 - 2017) is available at https://doi.org/10.5194/essd-2019-136."

A link with the Essential Oceanographic Variables (EOV) and Essential Biodiversity Variables (EBV) as recommended for a well-suited monitoring programme should have been interesting to develop.

We fully agree with the reviewer and this was an error

The materials and methods are described in sufficient details. The authors provided main characteristics of each sensor and also the associated range, accuracy, resolution and response time. The quality of the data is assured as the authors follow (well documented) good practises at sea and in lab. All these information are also included in the two associated SEANOE archives. We can however regret the absence of a validation protocol for fluorescence data.

Fluorescence sensor was calibrated following the instructions described by Sea-Bird Electronics Inc. in its APPLICATION NOTE NO. 9, downloadable at https://www.seabird.com/application-notes,

where SEASOFT Coefficients for the Sea Tech Fluorometer are calculated. However no validation of the data has been applied after the cruises.

Anyway the sentence was rewritten as follows, adding a new reference (see p.7 new lines 199-202): "Chl-*a* was calibrated following the instructions given by SBE (2008a) where the SEASOFT™ coefficients for the Sea Tech fluorometer are calculated. No validation of the data has been applied after the cruises. It is reported as Relative Fluorescence Unit (R.F.U.) in the datasets. Also pressure sensors were not calibrated as usually stable but, if problems occurred, it was sent to SBE Inc. in USA for its calibration."

Added the following reference in bibliography "SBE: AN-9: Calculating SEASOFT Coefficients for Sea Tech Fluorometer and WET Labs Flash Lamp Fluorometer (FLF). 1 p, 2008a"

I'm not sure that figures 2, 3 and 5 are very helpful.

We thought that figures could improve the description of the instruments or labs on the text but, as both reviewers noted a redundancy in description plus figures, we have deleted them. Then the numbers of the figures have been updated in the text.

To conclude, I propose that the manuscript should be **accepted subject to minor revisions**.

**Minor corrections and comments directly in the text as comments in the attached PDF file:**

New pages and lines are mentioned if they changed after correction as *new p.x and7or new line y*.

p.2, line 43: put *in-situ* in *italic* in the whole text. Here it disappeared due to the rewriting of the whole sentence.

p.2, line 44, now lines 43-44: whole sentence rewritten accordingly "Due to such an oceanographic importance, since the '50s French or Italian unsystematic cruises have been organized in the area whose data are available in the PANGAEA (https://www.pangaea.de) and SEANOE repositories (Dumas et al., 2018)."

p.2, line 46, now lines 45-46: acronym defined in the text. Sentence rewritten as "the Science and Technology Organisation - Centre for Maritime Research and Experimentation of the North Atlantic Treatment Organization (NATO STO-CMRE), based in La Spezia,"

p.2, line 53: sentence deleted as it did not improve the information on instruments or data then previously

p.3, line 69, now line 68: changed accordingly

p. 3, line 71, now line 70: I would leave the past "was" as this vessel does not exist anymore

p.3, line 83: changed in the whole text for all parameters

p. 5, line 130, now line 126: changed accordingly

p.5, line 130, now lines 126-132: added what requested but rewriting the sentences as follows "For Chlorophyll-*a* (Chl-*a*), CDOM and DOC all the water samples were filtered on 0.42µm Whatmann GF/F glass microfiber filters for Chl-*a* and 0.22µm membrane filter (Sartorius, Minisart, SM 16534 K)

for DOC and CDOM; this to remove the particulate fraction immediately after collection. Then filters were frozen at +4°C to be analyzed at labs following standard procedures described by Lazzara et al. (1990) for Chl-*a*, and by Vignudelli et al. (2004) and Santinelli et al. (2008) for CDOM and DOC. For nutrients, during the first three medgoos cruises water samples were partially analyzed on-board by a Systea μCHEM Auto-analyzer, while remaining samples frozen at -20°C following Strickland and Parsons (1972). In the following cruises all water samples were frozen and then analyzed once back."

p.5, line 135, now line 132: checked in the whole text and reformatted following the Copernicus Word Template

p. 6, line 168, now line 164: corrected

p.6, line 173, now line 168: corrected

p.6 line 175, now 170: corrected

p.7 line 180, now p.6 line 175: corrected

p.7 line 181, now p.6 line 176: corrected

p.7 line 201, now p.6 line 196: rewritten as "If the shift was random, Soc and Voffset for dissolved oxygen, slope and offset for conductivity/temperature were recalculated and then the data corrected following the procedures described in the application notes number 64-2 (SBE, 2008b, 2012) and 31 (SBE, 2016), respectively." Added the following references in bibliography " SBE: AN-64-2: AN64-2: SBE 43 Dissolved Oxygen Sensor Calibration and Data Corrections. 5 pp, 2012" and "SBE: AN-31: Computing Temperature & Conductivity Slope & Offset Correction Coefficients from Lab Calibration & Salinity Bottle Samples. 8 pp, 2016"

p.12 line 355: figure deleted

p.12 lines 356-357: figure deleted

p.12 line 359: figure deleted

p. 14 ex-figure 4, now figure 2: changed decimal separator accordingly

p. 15 ex-figure 6, now figure 3: changed decimal separator and depth format accordingly

p. 16 table 1: changed decimal data format according to ISO 8601

**Reviewer's 2 comments on the following manuscript:**

**essd-2020-17, Submitted on 10 March 2020**
**Quality assurance and control on hydrological data off western Sardinia (2000–2004), western Mediterranean.**
**by Alberto Ribotti, Roberto Sorgente, and Mireno Borghini**

**Reviewer**: Dr. Sarantis Sofianos (Referee)

Dear Dr. SOFIANOS,

we thank the you for having accepted the review and improved our manuscript with your comments and suggestions.

We agree with the points raised and we have answered all your comments.

Below, reviewer's comments are given in normal font and our responses in red.

The manuscript "Quality assurance and control on hydrological data off western Sardinia(2000 - 2004), western Mediterranean" by Ribotti et al. presents an open access dataset (and its quality control methods) that includes data acquired during seven oceanographic cruises off the coast of western Sardinia, Western Mediterranean. The main aim of the cruises and the publication of the dataset is to give to the scientific community a contribution to the local circulation and interaction with the general Mediterranean circulation. The description of instrumentation and quality assurance and control methods of the dataset is clear and informative (although the pictures of specific instruments seem redundant).

We thought that figures could improve the description of the instruments or labs on the text but, as both reviewers noted a redundancy in description plus figures, we have deleted them. Then the numbers of the figures have been updated in the text.

The information is also included in the SEANOE archives. Based on the aim of the cruises, the absence of hydrodynamic (ADCP) data makes the dataset "weaker" and less informative. The reason for not including this data should be better explained by the authors.

Thanks for having raised this point. As explained to referee 1 too, LADCP measurements at CNR started in those years and ADCPs were not available during all cruises. After their acquisition, usually current data were stored but never used. A great job to rescue, reprocess all raw data and build a complete high quality database is ongoing in these months.

The following sentence has been added in the 6th paragraph (p.7 lines 211): "The rescue and reprocessing of hydrodynamic data is ongoing in order to reorganize them in a quality controlled database."

p.1 line 15: corrected

p.1 line 28: corrected

p.2 line 35: corrected

p.2 line 44: apart from the sentence was completely rewritten

p. 3 lines 83-84, now lines 82-83: units corrected

p.4 line 96, now line 95: corrected

p.6 line 156, now line 151: corrected

[revised manuscript text omitted]

385    **Figure 1**

[Figure]

[Figure]

**Figure3**

[Figure]

395

**Figure4**

[Figure]

400

**Figure5**

[Figure]

**Figure6**

[Figure]

405

[Figure]

[Figure]

410

**Table 1**

| | medgoos1 | medgoos2 | medgoos3 | medgoos4 | medgoos5 | medgoos6 | medgoos7 |
|---|---|---|---|---|---|---|---|
| Date | 282000/05-02/28-2000/06/200002 | 232001/03-03/23-2001/04/200103 | 2001/09/ 10-20/09/2001 | 2002/05/4-23/05/2002 | 312002/10-18/31-2002/11/200218 | 282003/03-17/28-2003/04/200317 | 2004/01/07-26/01/2004 |
| # days | 6 | 12 | 11 | 20 | 20 | 21 | 20 |
| R/V | Urania | Urania | Urania | Urania | Urania | Urania | Urania |
| # CTDs | 38 | 67 | 41 | 68 | 42 | 92 | 87 |
| Reverse Thermometer (bottle #) | | | | 3 (1, 3, 5) | 4 (1, 3, 5, 7) | | |
| SBE-13 O2 Sensor | X | X | X | X | | | |
| SBE-43 O2 Sensor | | | | | X | X | X |
| Fluorescence | X | X | X | X | X | X | X |
| LADCP | X | X | X | X | X | | |
| SADCP | X | | | | | | |
| **CTD secondary sensors** | | | | | | | |
| Secondary Temperature | | X | X | X | X | X | X |
| Secondary Conductivity | | X | X | X | X | X | X |
| SBE-43 O2 | | | | | X | X | X |
| **Water samples for following analyses of** | | | | | | | |
| DOC | 23 | 66 | 39 | | 55 | | |
| Nutrients | 23 | 66 | 39 | | 55 | | |
| Phytoplancton | | X | X | X | X | | |
| Chl-*a* | | 66 | 39 | X | 55 | | |
| **On-board analyses** | | | | | | | |
| O2 | X | | X | X | X | X | X |
| Conductivity | X | | X | X | X | X | X |

415